# Developing Marsha and Marian's Neighbors: A Shared Housing Intervention to Address Homelessness among LGBTQ+ and Pregnant/Parenting Youth

**Maurice N. Gattis \*, M. Alex Wagaman and Aaron Kemmerer**

School of Social Work, Virginia Commonwealth University, Richmond, VA 23284-2027, USA
\* Correspondence: gattism@vcu.edu

**Abstract:** Background: The aim of this paper is to describe the development of a shared housing model intervention that was designed to serve youth experiencing homelessness who are LGBTQ+ and/or pregnant/parenting. The intervention is built around two guiding philosophies: housing first and restorative justice. Methods: We engaged in a year-long planning process with an advisory group from 1 July 2021 through 30 June 2022. The advisory group is a partnership between Virginia Commonwealth University School of Social Work, seven young people with lived experience expertise, and members from four partner organizations, including local organizations serving LGBTQ+ youth. Results: Marsha and Marian's Neighbors is designed to provide safe, comfortable housing for 12 months for 20 young people and their children if they have any. Participants will live in apartments where rent and utilities are paid for by the program for all participants. The program also provides money for arrears that may prevent the participants from being able to secure a lease, mental health appointments, legal consultations, and doula services provided by partners recruited by the planning team. Services are provided by staff and partners that are LGBTQ+ affirming and pregnant and parenting affirming. Conclusion: One of the most important lessons that we learned is the power of developing shared values in a novel intervention for LGBTQ+ youth. One focused intervention that supports both pregnant and parenting youth who identify as LGBTQ+ remains necessary. Paradoxically, LGBTQ+ people, particularly bisexual, lesbian, and queer cisgender women, are at an equal or greater risk of unplanned pregnancy. Both groups are vulnerable to housing instability independently, and those who live at the intersection of being LGBTQ+ and pregnant/parenting face an elevated risk for housing instability.

**Keywords:** homeless youth; parenting youth; LGBTQ+ youth; housing intervention

## 1. Introduction

The prevalence of homelessness among youth in the U.S. is staggering. Through their Voices of Youth Count point-in-time count, Chapin Hall at the University of Chicago reported that 1 in 10 young people (ages 18–25) and 1 in 30 adolescents (ages 13–17) experienced homelessness over a 12-month period [1]. In 2012, the United States Interagency Council on Homelessness (USICH) released public information about its strategic priorities to address homelessness among children and young people [2]. While definitions of 'youth' and 'young people' vary from culture to culture, we can rely on definitions from global-level organizations. The United Nations (UN) defines 'youth' as between the ages of 15–24, and the World Health Organization (WHO) defines adolescence as the developmental stage between 10–19 years old [3]. WHO also combines 'adolescence' and 'youth' into the category of 'young people,' those ages 10–24 years old [3]. The present paper and the specific intervention detailed focus on youth/young people from ages 18–24.

In extant literature about housing and homelessness, social work researchers have focused on prevention and intervention strategies for youth experiencing homelessness (YEH). Some of the intervention literature emerged as early as the 1990s, now fully three

decades ago [4–9]. Early interventions literature focused heavily on runaway youth, as well as substance misuse and HIV treatment for YEH. A robust body of literature is now available about interventions aimed at ameliorating youth homelessness and housing instability. Many have targeted 'risky' behaviors, such as survival sex and substance use [10–17]. Some interventions are specifically designed for YEH who were formerly in foster care [18–20]. Interventions for YEH include a wide range of approaches: supportive housing services such as shelter and transitional housing [21–23], mental health treatment [24,25], mentoring [26], mindfulness training [27], and employment programs [28]. These interventions largely either address what has been identified as individual causes of homelessness or the development of coping strategies as opposed to structural shifts.

### 1.1. Lesbian, Gay, Bisexual, Transgender, and Queer (LGBTQ+) YEH

Although most YEH benefit from supportive housing services, LGBTQ+ YEH require specifically tailored intervention strategies because they are more likely to experience discrimination, victimization, suicidality, substance use disorders, family rejection, and domestic violence compared to their cisgender heterosexual peers [29]. The Williams Institute reported that LGBTQ+ young people are disproportionately represented among the youth population accessing housing services, with researchers estimating that 30% of YEH identify as LGBTQ+ [30]. Transgender youth may be particularly vulnerable to homelessness, yet many face barriers to adequate and affirming care at supportive housing services [31]. A report from Chapin Hall's *Voices of Youth Count* states that LGBTQ+ youth are at over two times the risk for homelessness compared to their cisgender heterosexual peers [32].

Shelton et al. examined LGBTQ+ YEH across seven different U.S. cities and found that young LGBTQ+ people are often rejected for their sexual or gender identity and asked to leave from families of origin, foster families, or group homes [33]; this was especially true for transgender YEH. 'Natural' supports, such as a family of origin, may reject LGBTQ+ youth, although 'coming out' and experiencing family rejection is only one pathway to homelessness for LGBTQ+ YEH, with other causes, such as family poverty and generational housing insecurity, sometimes playing a role in the lives of LGBTQ+ YEH [30,32]. While not a formal intervention, community-based housing supports have existed in the LGBTQ+ community for a while, particularly among Black/African American LGBTQ+ people. Researchers can look at the frequency of reliance on chosen families, the history of the house ball scene in New York, and mutual aid for housing among LGBTQ+ people [34–36].

An emergent field of literature about interventions for LGBTQ+ youth specifically has come out in the past two decades, which compensates for the dearth of literature that existed before the turn of the 21st century. Abramovich (2016) described the macro, cyclical nature of the relations with homelessness interventions, wherein services are not affirming for LGBTQ+ YEH who continue to have unmet needs. Due to the oppressive nature of the shelter system, LGBTQ+ YEH avoid shelter/housing services, and the system remains unimproved. On a micro level, Abramovich [37] also described how misgendering of trans youth during the intake process could alienate them from the very start of the intervention, and placement into a gender-segregated facility can cause LGBTQ+ YEH to distrust the shelter system and service providers to care adequately for their needs. Maccio and Ferguson [38] researched significant gaps in services for LGBTQ+ YEH; a thematic analysis of interviews with 24 service providers identified salient gaps related to housing services, educational services, employment services, family services, LGBTQ+ affirming, and culturally competent care. Regarding the housing services gap, Maccio and Ferguson [38] recommended (1) more crisis beds allocated specifically for LGBTQ+ YEH, (2) permanent supportive living programs, and (3) housing options for older LGBTQ+ YEH. In a systematic review of the literature on youth housing interventions, Detlaff and colleagues [29] showed that multiple studies highlighted the need to focus on transgender youth and youth of color for intervention and prevention strategies. Their systematic review also highlighted that studies showed access to emergency shelter and supportive housing interventions is positively associated with stable housing outcomes post-discharge [29].

## 1.2. Pregnant and Parenting YEH

Separately from their reports on LGBTQ+ youth, Chapin Hall reported that pregnancy and parenting are common among YEH: 44% of young women and 18% of young men (ages 18–25) were pregnant or parenting at the time they were experiencing homelessness [1]. There may be a heightened risk for unplanned pregnancy among young adults experiencing homelessness who were/are also systems-involved youth. Young people with foster care or juvenile justice histories indicated a higher risk for engagement in trade sex and unplanned pregnancies compared to YEH, who were not systems-involved [39]. However, not all pregnancies among YEH are unplanned, and not all YEH view parenting as a negative experience. Despite challenges in securing the basic needs of themselves and their children, some YEH viewed parenting as an opportunity to step up into a role of responsibility and gave them a greater sense of purpose [40]. Some perceive the experience of pregnancy and parenting as a chance to demonstrate their positive parenting skills and provide better parenting than their own caregivers provided [41]. Still, the challenges faced by young people who are pregnant or parenting and homeless are immense-the lack of housing stability can affect access to prenatal care and social supports, which usually assist with child-rearing [41].

## 1.3. The Intersection of Both LGBTQ+ and Pregnant/Parenting Youth

There is not much literature about LGBTQ+ pregnancy and parenting specifically, and even less about YEH, who are both LGBTQ+ and pregnant/parenting. Extant literature indicates that paradoxically, cisgender LGB girls 15–20 years old are more likely than their heterosexual peers to become pregnant [42,43]. Additionally, LGBTQ+ parents face increased social marginalization and stigma compared to heterosexual, cisgender peers, though no findings show significant differences in behavioral, social, or educational outcomes when comparing children of LGBTQ+ parents to children of heterosexual parents [44]. In a society built around Judeo-Christian norms and mores, the dominant cisnormative and heteronormative ideology associates parenting with 'male/female' reproduction [45]. This hegemonic ideology perpetuates stigma and internalized stigma toward LGBTQ+ people who are pregnant or parenting. Alday-Modaca and Lay-Lisboa's [45] research showed that LGBTQ+ parents feared they and their children would be socially rejected based on this dominant ideology. LGBTQ+ parents also feared that their own children would be socially marginalized because they had a queer parent or feared that there was an assumption that the child themselves would identify as LGBTQ+ [45]. The anticipation of stigma relates to Tate and Patterson's [46] finding that lesbian and gay young people (ages 18–35) are less likely to aspire to parenthood than their heterosexual peers.

Indeed, the study of parenting itself is still limited within a cis/heteronormative context [47]. In the past decade, research has offered more insight into lesbian family structures and parenting but not as much about cisgender gay, bisexual, or transgender parents (though there is some recent literature emerging about these populations) [44]. The focus on lesbian parents aligns with the fact that most literature on pregnancy and parenting among YEH focuses on young women and mothers–under the cis/heteronormative ideology, the social construct of 'parent' typically equates with 'mother' [41]. Within this binary gendered paradigm, transgender parents may face additional struggles as parents [45]. The earliest literature on trans men birthing children goes back to only 2014, and the few research studies on transwomen as parents relate to their role as parents with cisgender women partners [47]. Healthcare systems, social service providers, and the law struggle to keep up with the 'changing face' of family formation-often 'invisibilizing' those parents who diverge from the traditional gender roles that are expected in a heteronormative context [47].

It remains important to note that LGBTQ+ family structures form in diverse ways (e.g., adoption/fostering, medical fertilization processes, using egg or sperm donors, single parents, and step-parenting). Just as cisgender heterosexual people parent children in various ways, there are also innumerable ways for LGBTQ+ people to become parents [44]. Researchers stress the importance of clinicians and health services providers' encourage-

ment and validation of LGBTQ+ parents; recognizing and showing support for queer families with children can improve the health of the parenting process. Clinicians and health care providers can offer an important source of social recognition of the LGBTQ+ family structure by treating parents with dignity and respect [48].

Among housing services providers and researchers, there is often an incorrect assumption that LGBTQ+ YEH will not be involved in romantic relationships that can produce a pregnancy, and the topic is generally not considered for this population. However, research indicates that LGBTQ+ YEH become pregnant or involved in pregnancy at equal or higher rates than their cis/heterosexual counterparts [41]. The literature documents that there are specific challenges for the youth belonging to all three distinct experiences of homelessness [49], pregnancy and/or parenting [1], and LGBTQ+ identity [29,30,37,38]. The circumstance of young people who are experiencing all three simultaneously has been less researched. A lack of research does not equate to the lack of existence of the phenomena, nor does it negate the challenges involved in the lived experience of LGBTQ+ YEH who are pregnant or parenting. In fact, the dearth of research in this area calls for more investigation and evaluation of effective interventions for YEH who are LGBTQ+ as well as pregnant and parenting.

### 1.4. Shared Housing

There are several different terms used for similar/the same phenomena: shared housing. These include collective living, co-housing, roommates, and multi-family housing. The planning team for this pilot intervention chose to utilize the term 'shared housing' to describe the program model. 'Shared housing' is defined as cohabitation in dwellings among people who are unrelated and who share the cost burden of the living space. For the younger generation, shared housing has become increasingly common, with few living-wage jobs and a lack of affordable housing [50]. Some people may feel apprehension about living with unknown others who are unrelated biologically or through intimate relationships. However, there is an economic push that has made the shared housing phenomenon more common and created new types of cohabitation relationships [51]. In the social work literature, the concept of shared housing emerged initially as a concept used to support elderly people-proposed as an alternative to nursing home care or 'aging-in-place,' creating an environment where seniors could pool limited resources and offer social support to each other [52,53]. Recent literature reviews have also looked at the ways that shared housing may improve health outcomes for elders more favorably than nursing home care [54].

In recent years, more scholars have been discussing the advantages (e.g., financial, health, and social benefits) and disadvantages (e.g., concern about financial reliability of housemates, lack of privacy, and interpersonal conflict) of shared housing for young people specifically [50,55]. Public health researchers have recommended that shared housing arrangements can increase social support and financial stability [56].

Shared housing involves sharing space and sharing time. Practiced in reality, the aspects of shared housing go beyond just the physical shared space but also involve establishing and maintaining interpersonal relationships by spending time together [57]. The social aspect of shared housing is crucial for young people to build interpersonal skills and supportive peer networks, especially for youth who are LGBTQ+ and/or pregnant and parenting-there is great power in shared identities among roommates or supportive others. Shared housing can be a tool for collective empowerment and social support [58].

### 1.5. Guiding Frameworks

In the United States, homeless services have significantly shifted in their approach since the turn of the century. Previously, programs and services were driven by an assumption that people experiencing homelessness needed to show 'readiness' for housing, often being asked to address individual challenges before being connected with a permanent housing placement. The shift toward what has been labeled a Housing First approach aims to move

people quickly toward housing, prioritizing it over other areas in which a person might be facing barriers. This has been particularly relevant for people with mental illness and substance use, two populations who were often limited or delayed from accessing housing and given unrealistic timelines and goals. Housing First is rooted in a human rights framework, recognizing that housing is a human right to which everyone should have access rather than proving 'worthiness' or 'readiness.' Similarly, Housing First honors choice and control instead of mandating engagement or compliance with services in order to access housing. Instead, programs taking a Housing First approach offer supports and services, honoring the client's right to choose what they do or do not want to receive or engage.

Aligned and interconnected with a Housing First model is a restorative justice (RJ) approach, the values of which undergird a harm reduction approach. Harm reduction and RJ both recognize that people engage in a variety of coping strategies to deal with the impact of trauma. Rather than penalizing or criminalizing those coping strategies, RJ and harm reduction suggest that we (1) recognize them for what they are, (2) honor that people are doing what they need to do to survive, and (3) support them in minimizing the impact of those coping strategies or behaviors. RJ adds a layer of knowledge to this approach that the systems and institutions in our society are the sources of trauma. Instead of individualizing and thus blaming people for their trauma, which often happens in the case of homelessness; for example, RJ calls us to recognize homelessness as trauma and to acknowledge the systems and structures that would allow a person to be without housing, a basic human need. RJ, and its adapted predecessor, transformative justice, invite us to consider the ways in which we hold space for people as they are and create spaces and opportunities for healing in community with others.

Given the context within which LGBTQ+ and/or pregnant/parenting youth in the U.S. are operating and what is required to navigate homelessness as previously described while holding the possibility of the incorporation of approaches such as Housing First and restorative justice, these approaches and their guiding principles were centered in the development of the program model described herein. The model being developed fills gaps in service provision and the literature regarding interventions designed specifically for LGBTQ+ youth, pregnant and parenting youth, and youth at the intersection of the population using restorative justice and housing first principles.

## 2. Material and Methods

### 2.1. Marsha and Marian's Neighbors

The main aim of this paper is to describe a shared housing model program that was designed to serve YEH who are LGBTQ+ and/or pregnant/parenting. The present program under evaluation is built around two guiding philosophies: housing first and restorative justice. Firstly, embracing a housing first philosophy involves providing quick access to safe and stable housing with no preconditions (like sobriety or treatment) while also offering services and supports that may help the person who needs housing [59]. Secondly, restorative justice (RJ) focuses on healing instead of punishment–core principles of RJ include accountability for harm, centering the victim/survivor in the process, and respect for all people involved [60].

Aligned with a housing first philosophy, the main activity of the program is financial assistance, with robust case management and health/mental health support. This program intends to pay participants' rent every month for 12 months, thus alleviating the stress of financial instability for young people in the program. Previous practice experience and feedback from YEH in the local community have shown that rapid re-housing (supporting a household on one's own) does not work for all young people. Initially, the program design called for participants to live alone or with roommates. Over time, during the planning process, the planning team decided on a roommate matching process for all participants. This process can help build peer support networks and facilitate permanent connections among young people on their journey to housing stability. The program also emphasizes

collaboration with community partners who serve LGBTQ+ YEH, pregnant and parenting YEH, and youth living at the intersection of all three experiences.

As per grant funding requirements, youth involved in the program will fulfill monthly sessions with a case manager, but all other services will be voluntary. The monthly case management sessions (at a minimum, as required by the funding source) include recertification of financial eligibility for the program and the development and monitoring of a housing plan, which is collaboratively developed and centers on goals that the young person wants to work on with the intention of enhancing capacity for maintaining permanent housing once their program participation ends. These are highly individualized. The program will have low/no barrier to program participation, i.e., youth will not be required to be sober or have formal documentation to participate. Additional support for YEH in the program will include mental health services, doula services, legal aid services, peer navigation, case management, and weekly to biweekly programs and events. Partners who provide these services are LGBTQ+ affirming. The team will take steps to evaluate the effectiveness of the program by measuring youth outcomes and the ability to replicate the program in other areas throughout the state. The program aims to serve 20 young people by the end of the 12-month period. With programmatic support, we predict that YEH will experience improved housing and financial stability, achievement of educational and vocational goals, and increased well-being and permanent connections.

### 2.2. Local Community Context

Community and sociopolitical contexts are important factors for understanding the design, implementation, and ultimate outcomes of a program such as this. Design and implementation of the program model described herein took place in a midsize city in the Southeastern United States. Models for housing programs designed to meet the needs of young people must be adapted to the community within which they will be implemented, particularly given the way in which factors such as housing policy, community infrastructure, and prioritization of youth in existing homeless services impacts what is required to adequately meet the needs of young people. Similarly, the socio-political time within which the program was imagined and developed has shaped key aspects of the design.

Historically, the community within which Marsha and Marian's Neighbors was designed has not had a focus on youth as a population experiencing homelessness that requires programs and services tailored to meet their needs. Similarly, the community has not had the capacity or infrastructure to meet the unique needs of LGBTQ+ people experiencing homelessness. As with many systems, the approach to address these gaps has largely been additive-incrementally changing the existing system to add components that might address unmet needs rather than building something from the ground up that centers the needs of the target populations. Beginning in 2014, an advocacy group largely made up of youth and young adults with lived experience expertise began documenting through research the existence, prevalence, and unique needs of youth in the community. The research conducted by this youth-led local advocacy group has been used to engage in awareness-raising, education, and advocacy efforts intended to increase the visibility of youth homelessness, gain acknowledgment of the need for a community response that is youth-specific and youth-directed, and lead efforts to design and deliver these programs and services. In 2019, with the support of a national organization aimed at building a movement to end youth homelessness, core partners came together to try to increase the ways that the local homeless services system works in a youth-affirming way. The partners included local LGBTQ+-serving organizations, the primary youth-serving homeless services organizations, a homeless services/health outreach organization, university social work professors, and representatives from the youth-led local advocacy group. Through this work, a broader and deeper understanding of the existing system was developed, including key gaps in services. In addition, trusting relationships were formed. The members of the team learned how to talk with and take action together in ways that fostered an interest in future partnerships rooted in a core set of values.

In the Spring of 2020, the pandemic hit, and community resources shifted to manage a crisis that prioritized limiting the transmission of COVID-19. While the financial response to the pandemic at the federal level allowed for temporary rental assistance and other forms of financial support to individuals, it did not address the severely limited affordable housing stock for renters with the lowest incomes. Simultaneously, housing costs continued to increase, and housing stock continued to decrease. This economic context created a situation in which many youths were entering into leases at rent levels they could not sustain, often resulting in evictions. Given the competitive housing market, the ability for these youth to then secure housing post-eviction was nearly impossible.

Early in the pandemic, providers identified an increase in two populations of youth presenting for emergency homeless services: LGBTQ+ youth and pregnant or parenting youth, including youth at the intersection of these populations. Though the existing homeless services system had limited data on these populations, other forms of data from providers working on the streets and LGBTQ+ organizations doing research and crisis response helped to paint a picture of the need. The increase in youth among these populations presenting for services was brought to the previously described group of partners, and a plan was made to pursue funding through a state-designated funding source. Given the context as previously described, a shared housing model that was tailored to the unique needs of LGBTQ+ youth, pregnant and parenting youth, and youth at the intersection appeared to be an opportunity to navigate existing barriers and constraints and work in a youth-affirming way. The funding was awarded and provided resources to engage in a year-long planning process.

What follows is a description of the process and components of the resulting design of Marsha and Marian's Neighbors, named for community ancestor Marsha P Johnson (queer/trans liberation movement) and living legend Marian Wright Edelman (Children's Defense Fund). Through this paper, we hope that other communities can replicate or adapt what we have done in order to meet the needs of underserved populations of youth experiencing homelessness.

### 2.3. The Program Planning Process

We engaged in a year-long planning process with an advisory group from 1 July 2021 through 30 June 2022. The advisory group consisted of a partnership between two full-time associate professors at Virginia Commonwealth University School of Social Work, seven young people with lived experience expertise, and members from four partner organizations, including local organizations serving LGBTQ+ youth and YEH. More than 51% of the group were young people with lived experience of homelessness. The youth-centered organization that provides support services to LGBTQ+ youth and young adults has stated its commitment to black LGBTQ+ young people in the community. They regularly provide youth support groups, counseling services, a library, a binder exchange program, hygiene products, and lockers, and have drop-in hours where people ages 11–25 can talk with staff, receive a meal, and/or enjoy the space.

A second community partner is dedicated to serving families facing homelessness, autism, developmental disabilities, mental illness, and special needs education. Staff members who work in their homelessness department served on the planning committee and offered their expertise using their experience with their rapid re-housing services, housing resource center, youth outreach services, and their section 8 housing community. Another partner is a Black trans-led organization that provides local LGBTQIA+ communities with access to HIV and STI testing, linkages to care, prevention, and advocacy. They engage in street outreach and club outreach and have a food pantry, host game nights, a support group for transgender women, emergency housing assistance, peer support for people living with HIV, a support group for trans men, a computer lab, and a support group for non-binary, gender non-conforming and questioning trans individuals.

Our final community partner is a local advocacy group and a youth participatory action research team affiliated with the University School of Social Work, comprised of

youth with lived experience expertise in housing instability and homelessness. They engage in data collection and dissemination of their research findings, including in peer-reviewed academic journals. An important component of their work is advocacy around issues that emerge in their research findings. The majority of the members of the participatory action research team are youth of color, LGBTQIA+ youth, and LGBTQIA+ youth of color.

Young people were recruited through partner organizations and local housing programs and also represent a range of distances from housing instability, which added an element of peer support.

The advisory board met bi-weekly virtually, and there were two in-person half-day retreats. Each participant who was not serving on the board as a service provider received $30 per hour, and the partner organizations received $5000 compensation for the participation of their staff member for one year. The meetings began with a check-in question and ended with a check-out question to create an environment where team building and relationships were as important as the task at hand. The group also identified learning edges which were addressed by having guest speakers give presentations at the meetings. Two members of the advisory board conducted two focus groups with currently unhoused young people to get feedback that informed the development of the program. Individual meetings with partners to gather feedback were scheduled to get specific input based on the expertise of the partner. An important outcome of the work of the advisory group was the development of a shared housing pledge (see Appendix A) which is a statement of values that was developed and served as a guide to the collective work of the program staff, advisory board members, and advisory board.

### 2.4. Values-Based Model

As the planning team for Marsha and Marian's Neighbors began working on designing the program model, it became clear that a set of shared values and a commitment to uphold those values would need to be developed and then used to guide the rest of the design process. Young people shared stories of the harm that had been done when, for example, they sought services from an organization that made declarations of being LGBTQ-affirming but did not practice in a way that aligned with this declaration. Similarly, providers from our partner organizations shared the damage done when they fear harm due to a referral they make for a youth. These experiences, absent a shared commitment to values and a process for naming and repairing harm, made it difficult for the planning team to imagine something different. So, the planning team spent time delving into questions such as, 'What does it look like in action to be LGBTQ+ affirming?' These conversations informed the development of a pledge that was a manifestation of what the values look like in action and in relationship with one another. The program pledge (see Appendix A) is to be used for all stakeholders engaged in or with the program, including participants, staff, advisory board members, and community partners. Moreover, it created a shared set of commitments that the planning team could look to in times of uncertainty or differing opinions to guide the work of designing the program.

The values are rooted in a basic understanding that a Housing First approach is intended to reduce barriers to accessing housing, which is informed by a human rights framework. If we view housing as a basic human right, then our approach will be rooted in doing what is necessary to get people housed rather than having people prove they are 'ready for' or 'worthy of' housing. Housing First also requires that we take a harm-reduction approach to meeting youth where they are. Rather than identifying behaviors or coping strategies that will eliminate their participation, we work to identify ways to include youth and reduce the harm that may result from those behaviors. Then, we commit to working hard to create spaces where they no longer need the coping strategies that can cause them and others harm.

Community is another primary value that the planning team committed themselves to and incorporated into the design of the program. We know that community, and the social networks we establish, are essential to our ability to meet our basic needs, including

housing. This value is connected to those that come from a transformative justice approach, which incorporates processes of naming harm, working to repair harm when possible, creating and maintaining healing spaces, and engaging in accountability practices at all levels. These support the development and sustenance of community connections that become long-term supports for young people experiencing homelessness, which have been identified in the literature and system change work as permanent connections.

Once the pledge was developed, and the values had been clearly articulated, it became more clear what pieces of the program model were needed. For example, it was important to have a clear grievance process for young people in the program to name harm and to know how the staff and community of participants would be accountable for addressing it. A roommate matching process needed to make space for people to identify the coping strategies that they need a roommate to be comfortable with, as well as those that would make them feel unsafe in their housing. Roommate agreements incorporated ways to talk about how to reduce the chance of structural violence in a crisis. These were all ways that the values were reflected in the design.

As we approached community partners to share the pledge, we received positive feedback. Partner organizations appreciated the clarity of values, the ability to find alignment, and the upfront conversations about how to hold one another accountable to uphold the values in a spirit of valuing growth.

## 3. Results

### 3.1. Program Structure

The evolution of the program is an ongoing process that will continue to improve as program participants engage in the program and data is collected throughout the program evaluation. Overall, the program is designed to provide safe, comfortable housing for 12 months for 20 young people (ages 18–24) and their children, if they have any. While not all participants will identify as LGBTQ+, the program aims to provide an LGBTQ+-affirming environment, centering the unique needs and experiences of LGBTQ+ youth. Participants will live in apartments secured from the private real estate market where rent and utilities are paid for by the program for all participants. The program also provides money for arrears that may prevent the participants from being able to secure a lease, mental health appointments, legal consultations, and doula services provided by partners recruited by the planning team.

Recruitment for the program will happen in several ways, with attention to equal access. Young people involved with the planning team expressed concern that other programs they were familiar with required an agency referral to access housing programs or other resources. Given what we know about the reasons that youth, LGBTQ+ youth, and parents may not want to engage with services, we will allow a pathway for self-referral that will be made readily accessible. Self-referrals and agency referrals will open at the same time. The referral/interest form is brief, easy, and only collects information necessary to determine eligibility. Enrollment will be on a first-come, first-served basis, assuming that each interested person meets eligibility criteria (age, homelessness, LGBTQ+ or pregnant/parenting, or both).

There was a conscious decision not to create a waitlist for the program because the goal is for all participants to successfully complete the program and transition into an independent living situation, ideally in the same apartment that they lived in throughout the program. Creating a waiting list would potentially signal that we do not expect successful completion and may provide false hope to people on the waiting list while knowing that the likelihood of them obtaining an apartment in the program would not be likely given the commitment to program completion.

Client engagement will begin with an initial intake where there is an attempt to provide low-barrier access. Staff will conduct an urgency assessment which will allow them to help get potential participants safe and housed as quickly as possible. This process can also include working with partners who provide emergency shelter arrangements which

may allow for a more thorough needs assessment for supportive services. Roommates are determined using a holistic roommate-matching process developed by the planning team.

A variety of supportive services are incorporated into the program model that attempts to address the potential needs and wants of participants and is provided by program staff and community partners who also sign the pledge. A local organization that supports pregnant people serves as a partner that will provide full spectrum doula services, reproductive care, and abortion support. Similarly, providers who are LGBTQ+-affirming will provide mental health services in a community setting for program participants. Program participants will engage in monthly activities designed to address parenting, life skills, civic engagement, financial knowledge, wellness, art, and other activities identified by program participants and staff.

The staff for the program includes three full-time staff members: a program specialist, a peer navigator, and a resource advocate. The program specialist is responsible for overseeing a caseload of five participants, supervising two full-time staff, a B.S.W. intern, an M.S.W. intern, and fiscal oversight and administration of the program. The peer navigator assists all clients in activities during which participants may want a supportive person to be present with them, such as an important appointment. The resource advocate provides comprehensive case management to 15 participants in the program. All program staff help to develop and implement regular workshops for program participants. Two full-time university faculty members who dedicate approximately 10–15% of their time supervise the program specialist and the research and evaluation team, which comprises two Ph.D. students and two undergraduate research assistants.

### 3.2. Monitoring and Evaluation

An important component of the shared housing model is program monitoring and evaluation. The evaluation team aims to understand the challenges and successes of a shared housing model, as well as its effectiveness as an intervention in other local contexts. The monitoring and evaluation team will gather qualitative and administrative data about the program. Rigorous research has been integral to the entire process of program planning and implementation of this shared housing intervention, from holding focus groups to utilizing peer-reviewed literature to inform decision-making. Community-engaged research has informed the strategy of the project from its inception. Indeed, the foundational roots of the project are in community-based participatory research led by a council of young people with lived experience with housing instability (i.e., Advocates for Richmond Youth). Many of the community members involved in the formation of this project have been participating in community-based participatory research about youth homelessness in this local context. For this particular shared housing intervention, members of the planning team co-facilitated focus groups with young people. The results of these focus groups shaped program design and structure.

Conducting the research outlined in a monitoring and evaluation (M&E) plan will help us not only to improve program functioning but also to fund the program and expand it. To fund both this intervention program and the research process, the team recognized the importance of multiple funding streams: the main source of funds may not help us provide all we wanted. The project was a collaborative endeavor between the university and the community. However, maintaining both public and private sources of funding give the planning team and staff greater flexibility. For example, mental health practitioners reported that many young people have trauma from therapy and systems involvement, such as child welfare and criminal legal systems. With typical compensation structures, practitioners tend to offer traditional talk therapy. With private funding, we can provide diverse offerings, including seminars related to diverse topics such as cooking, arts-based healing, and trauma-informed care. Research findings will serve as a grant report for one funder, Virginia's Department of Housing and Community Development (DHCD), to evaluate the program's successes and challenges.

The practice of regular full team meetings (including principal investigators, planning team members, program staff, and research team members) helped to build relationships and cohesion on the team. During these full team meetings, M&E researchers were able to communicate about research plans and receive feedback from other team members (for example, researchers created a survey, which program staff reviewed and provided feedback, which was then incorporated in a second draft of the survey). This spirit of collaboration led to more refined research processes, where research design incorporated feedback from the staff and planning team. We were able to discuss what worked and what did not work, particularly prioritizing youth experiences: assessing emotional risk or practical concerns for participants involved in research design processes, such as 'Will this item trigger a participant?' This feedback from staff ensured that the research design assessed the outcomes needed without causing undue harm. The team consistently attempted to prioritize youth experiences in the program and focus the research without it dominating the team's agenda.

The M&E team will measure youth outcomes in key outcome areas to evaluate the program and report back to the community and our funders. Our main aim is to assess whether program participation is associated with psychosocial outcomes for youth, including (1) maintaining stable housing, (2) developing a peer support network, (3) building a sense of community, (4) increasing life skills, (5) improving mental health outcomes, (6) parenting, (7) critical action, (8) political participation, (9) social stigma regarding homelessness, (10) financial knowledge, and (11) resilience.

During three in-depth interviews (one at intake, one midway through the program, and one at program completion), researchers will ask program participants quantitative survey questions and open-ended qualitative questions to measure these outcomes.

The survey was developed using psychometrically validated scales and reviewed by the entire planning team using consensus to make adjustments according to team feedback. All participants will be informed about the risks and benefits involved in research participation. Signed informed consent paperwork will be filed for each participant. Participation in the research aspects of the program will be completely voluntary. Participants will receive an incentive to participate in the M&E interviews. Cash incentives are important to offer to participants as it validates that their time and sharing of their experiences remain valued by staff and researchers. Overall, the M&E plan emphasizes how program activities can facilitate participant progress in five key outcome areas.

## 4. Conclusion

*Alchemy: Lessons Learned in the Process*

Engaging in a year-long process to develop a novel intervention for LGBTQ+ and pregnant and parenting youth was revealed in a way that has implications for the development of services. One of the most important lessons that we learned is the power of shared values. As we engaged with members of the advisory board and cultivated relationships with potential community partners, they expressed appreciation for the opportunity to participate in a process that was explicit about its values and invited them to actively participate in the development of those values. As conflicts emerged, having a shared value system served as a consistent touchstone to help navigate the conflict and was a powerful tool in developing and now implementing the program.

Housing First does not mean housing only. As we engaged in the planning process, it became clearer that additional services are essential to a housing program. Services for the specific population that are the focus of the intervention need to have input from members of the population in order to make sure they are relevant and will be utilized. Having focus groups and survey data work to ensure that feedback on potential services is helpful in refining initial ideas about programs and services.

When developing services, working with partners is essential. A critical component that led to the successful completion of the planning process was a mix of direct service providers and young people with lived experience expertise in conversation. This also

required additional support to ensure that they could continue to participate in meetings, for example, compensation, connection to resources, food, and an affirming environment where they could see their voice and recommendations being incorporated into documents, processes, and procedures.

There were some cases where direct providers were not accommodated or compensated by their employer for participation, so we had separate meetings with them. This provided a little tension when trying to balance the needs of the young people and the needs of the direct providers regarding meeting times and other ways of contributing. The tension was worth navigating because relationships matter, and this cannot be done in isolation.

Balancing the needs of staff and youth in the program also emerged as a particularly salient issue in the planning process. Staff capacity is central to this idea-thinking about the importance of responding quickly/with urgency when youth are currently experiencing houselessness and taking the time to move with purpose and intention-finding a balance. One kind of program model cannot meet all of the needs of youth in a community. Being clear about what the program is intended to do and for whom is really important. This can also be challenging in a community where there are few/limited current services and programs for youth. For example, thinking about how our planning team wanted to be everything, a drop-in, provide case management even for youth not in our housing, etc. Considering what can realistically be done and by whom really guided the final model: Marsha and Marian's Neighbors.

In this context, there are several important limitations to the design. The age inclusion criteria do not adequately address the range of youth who are experiencing homelessness. Many young people that we work with advocate for increasing the age range of whom we can serve as youth, as the transition to adulthood is difficult, especially for marginalized populations. In addition, there are people under the age of 18 who could also benefit from the services. We also know that ideally, housing interventions should last more than a year to be effective. Our funding provides a year-long time limit, and we plan to have robust case management and resource support in place to provide a smooth transition into permanent housing after the program ends, and the design is that the housing would be the same housing they are living in during the program. One focused intervention that supports both pregnant and parenting youth who identify as LGBTQ+ remains necessary. Paradoxically, LGBTQ+ people, particularly bisexual, lesbian, and queer cisgender women, are at an equal or greater risk of unplanned pregnancy [61]. Both groups are vulnerable to housing instability independently, and those who live at the intersection of being LGBTQ+ and pregnant/parenting face an elevated risk for housing instability [62]. Trans and non-binary people are consistently erased from perinatal care [63], which highlights the need for an LGBTQ+-affirming program for pregnant and parenting youth.

While our program is derived from local data, anecdotal evidence, and community involvement, it addresses a problem that national data highlights. LGBTQ+ youth are overrepresented among youth homeless populations, and they face heightened risk while living on the streets [64–66]. Most mainstream services are adequate to address this-Marsha and Marian's Neighbors is designed to fill this important gap using Housing First and restorative justice approaches.

**Author Contributions:** Conceptualization, M.N.G., M.A.W. and A.K.; Methodology, M.N.G. and M.A.W.; Writing—original draft, M.N.G., M.A.W. and A.K.; Project administration, M.N.G. and M.A.W.; Funding acquisition, M.N.G. and M.A.W. All authors have read and agreed to the published version of the manuscript.

**Funding:** This research was funded by Virginia Department of Housing and Community Development, grant number: 21-HTF-031.

**Institutional Review Board Statement:** Not applicable.

**Informed Consent Statement:** Not applicable.

**Data Availability Statement:** No new data was created.

**Acknowledgments:** The authors would like to acknowledge Destini Barnette, Kimberly Embe, Michael Gifford, DeShaundra Otey-Easley, Caden Haney, Bri Mae Magsarili (they/them), Ariya Sharif, Anais Wyche (she/they), and additional members of the planning team who tirelessly devoted time and ideas to the planning process for Marsha and Marian's Neighbors.

**Conflicts of Interest:** The authors declare no conflict of interest.

## Appendix A

Marsha & Marian's Neighbors

Program Pledge

Through this pledge, all participants, staff, partners and stakeholders stand together to end youth homelessness by affirming the shared values in which this project is rooted. This pledge relies on your participation, skills and expertise, but most importantly your active voice in shaping how this program will shift to better serve our community.

I am a (select from the drop down) Stakeholder/Partner and I pledge to:

On a personal level:

- Be honest about what I know and learn what I don't
- Be aware of my own privilege and how it impacts others
- Acknowledge that I am a member of a larger community and I am accountable to the members of my community
- Understand my personal boundaries and recognize other's boundaries
- Take ownership of my role in this program and contribute to the democratic process of periodic program process/policy review and make suggestions for improvement
- Commit to be accountable for harm done to marginalized populations in a manner rooted in principles of community care and justice (Ex: acknowledge harm done, educate yourself about marginalization and affirm the harsh realities others face, approach healing with humility, recognize the power of storytelling, accept ambiguity of healing etc.)
- On a programmatic level:
- Center youth experience
- Commit to learning from others
- Honor folks with lived experience
- Give people grace and assume folks are doing the best they can with what they've got
- Understand people have different needs (no one-size-fits-all)
- Make space for people to show up as they are to get what they need
- Engage in transparent communication and recognize personal capacity
- Not give false hope, don't make promises that I cannot keep
- Empower youth to access low barrier services on their terms
- Be affirming of all identities, particularly LGBTQ+, pregnant and parenting, racially minoritized youth, disabled youth (and be willing to participate in required program trainings on these topics)
- On an organizational level among partners:
- To support staff representatives in getting the resources, training and support needed to uphold the pledge.
- To acknowledge organizational learning edges related to upholding the pledge and limitations of the programs they are a part of.
- To strive to increase organizational capacity for supporting staff in upholding the pledge.

As a program, Marsha and Marian's Neighbors will continue to advocate for systems and a community that aligns with the values of this pledge, in order to serve youth and young adults experiencing homelessness in ways that prioritize safety, healing and transformation.

As a pledger, I have had an opportunity to ask questions to clarify my role and what is expected of me. In the shared interests of the community I am pledging to join, I have reflected upon my needs and identified the following will help me to fulfill my pledge:

(Example: Training in LGBTQ+ affirming practice, an accountability partner etc.)

_________________________________________________________________________________
_________________________________________________________________________________
_________________________________________________________

I hereby agree to the above terms of my pledge and promise to uphold these principles for the duration of my connection to this project.

Printed Name: ___________________________________

Signature: ______________________________________

Date:___________________________________________

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
