# Peer review of "Developing Marsha and Marian’s Neighbors: A Shared Housing Intervention to Address Homelessness among LGBTQ+ and Pregnant/Parenting Youth"

_2673-995X, doi:10.3390/youth3010022_

Round 1

Reviewer 1 Report

Review of “Developing Marsha and Marian’s Neighbors: A shared housing intervention to address homelessness among LGBTQ+ and 3 pregnant/parenting youth

Purpose and Issues: This paper outlines the need for and development of a housing intervention aimed at supporting LGBTQ+ and pregnant/parenting YEH. The authors do an excellent job of setting up the importance of this housing program. I also appreciate the description of the collaborative team and the intended processes of the program. There are still details lacking, however, in the final takeaways from the development and future planning.

Abstract

1.      It is unclear how the housing intervention would uniquely support both LGBTQ+ young people and/or pregnant/parenting youth. Also, why combine these two groups of young people in one focused intervention?

2.     In the Conclusion sentence you don’t state the power of working with pregnant/parenting youth as well as LGBTQ+ youth and how this shaped the development.

Introduction and Background

1.     Strong introductory paragraphs clearly outlining the scope of the issue and the importance of continuing to assess the needs of YEH and interventions aimed at their wellbeing.

2.     Before moving into the background sections, I would like a clear 2-3 sentences on how your filling gaps in the literature with your study and why the intervention you’re developing is novel and significant.

3.     I really appreciate your clear and detailed emphasis on why LGBTQ+ and pregnant and parenting YEH face distinctive challenges that warrant tailored services and interventions. The next section then elaborates on the intersection between the two groups and their experiences – well done.

4.     Again, please conclude the Background sections with a clear statement(s) on your own study and how you’re contributing to these fields.

5.     Since you’re guiding principles are housing first and restorative justice, I would like to see these elaborated on in the Background or a Theory section. Right now they seem tacked on rather than really grappled with in understanding how these concepts are shaping your intervention development.

Methodology

1.      What do the “monthly sessions with a case manager” involve for youth?

2.     Very strong local and community contextual details!

3.     Excellent name and historically centered.

4.     Your description of all the collaborators, their affiliations, and how they were recruited and compensated is very well done.

5.     I assume the Appendix A would normally come at the end of the paper per standard Appendix placement?

Findings

1.     Will participants all identify as LGBTQ+ or is this more of a value/affirming-based approach of the intervention rather than an eligibility criteria? I imagine so since the goal is to minimize barriers of inclusion.

2.     How will young people apply to the program? I’m sure you’ll have to turn youth away if you only have 20 slots available.

3.     But how will you avoid a waitlist if you have limited available units? How will you enroll youth and decide who joins and what if there is more interest than available slots?

4.     Will the surveys and interviews with participants be voluntary as stated earlier that only the monthly sessions are required? Ok, now I see that they are voluntary and paid. And what if youth don’t participant in those monthly sessions, what are the consequences?

Discussion

1.     Your Conclusion is strong and reflective, but I would like more synthesis with scholarly research and how you’re positioning your findings/takeaways with other interventions in the field, especially in the realm of housing-first and RJ.

2.     You also need to expand more on the Limitations and challenges of the process, how you addressed them, and what you recommend for the future.

3.     More details on the future of the program is needed – when will it be implemented and will you follow up on the findings?

Reviewer 2 Report

Comments

For the fourth consecutive year, the number of homeless people in the US has increased: 580,000, according to an official report that does not cover the impact of the pandemic. African-Americans and Hispanics are most affected. According to a report published by the Department of Housing and Urban Development (HUD), between 2019 and 2020 the homeless population increased by another 2% and, “in a single night in 2020, approximately 580,000 people were experiencing the situation of homelessness”. housing» (https://pt.kamiltaylan.blog/us-department-housing-urban-development-hud/) . Of the 580,466 homeless people in the US, 39% are African American (although they make up only 12% of the US population). There is an increase of nine percentage points compared to that recorded in the previous year's report. The country's Hispanic population makes up 23% of those without coverage (and 16% of the total US population), according to HUD statistics, which recorded a 5% increase in homeless «Latinos» compared to the previous year's data.

For the first time since this statistic was started, in 2007, the number of homeless families with dependent children grew, comprising around 172 thousand people. If the majority of household members under the age of 18 were in protected sites (90%), the number of affected households that were not in a protected site increased by 13%.

The authors' approach does not address this reality. When the authors say”Although most YEH benefit from supportive housing services, LGBTQ+ YEH require 53 specifically tailored intervention strategies because they are more likely to experience dis-54 crimination, victimization, suicidality, substance use disorders, family rejection, and do-55 mestic violence, compared to their cisgender heterosexual peers”, this is a complex relative to “normal couples”. This problem has always existed in human history. Is this a situation that should be studied? Yes, but without the relevance and trend that the article wants to give you. I repeat, it has always been like this throughout history.

In 2021, California, with 161,548 homeless people (28% of the national total), is the worst affected state. New York (91,271; 16%), Florida (27,487; 5%) and Texas (27,229; 5%) follow. In these four states, cases of more than half of the homeless people in the USA are registered. In a statement, the National Association for the End of Homelessness highlighted the "significant" increase in people with some type of disability who experience "a chronic lack of housing and destitute lives", which rose by 15% between 2019 and 2020, "an indicator which suggests an increase in needs and vulnerabilities among the homeless population», says EFE. While acknowledging that the US federal administration invested "historical" funds in favor of the homeless during the pandemic, the association stressed that these numbers show how these "investments are tragically late" (https://pt.kamiltaylan.blog/us-department-housing-urban-development-hud/).

Last year, 61% of the homeless stayed in protected places and in emergency shelters or resorted to transitional housing programs, while 39% were in "places without shelter, such as the street, in abandoned buildings or in other places not suitable for human habitation», adds the document, quoted by the source. HUD secretary Marcia Fudge, who classified the numbers as "devastating", stated that the country has a "moral responsibility to end the housing shortage" and said she was aware that "the pandemic has only aggravated the crisis of people without home". The numbers that appear in this report are from January 2020, that is, from a phase in which the Covid-19 pandemic had not yet had the strong human and economic impact that it would have on the country.

Suggestions

The authors did not separate this reality from what happens with regard to those who belong to the LGBTQ+ group) YEH. Do they have to be “tailored intervention strategies”? (54). Why? Where is the explanation? This need more careful approach. I accepted the LGBTQ+ youth. But does the risk of pregnancy only exist in this group of people? Authors have to be more specific in their approach to what they upheld. This proposition is recognized by the authors themselves when they say that “There is not much literature about LGBTQ+ pregnancy and parenting specifically” (115).

Why should children, adopted or not, have gay or lesbian complexes? The rush to place this theme of the article and other literature on the agenda ended up stigmatizing this veracity in the negative. There is a national network in each country of public safety operators for lesbians, gays, bisexuals, transsexuals, transvestites and intersex people. This phobia is the set of practices, processes of violence and discrimination directed against a person because of their gender identity and/or their sexual orientation. Housing has little to do with this approach.

The authors should state that the neighbourhood has the propensity to treat and respect children and not individuals of the same sex. I do not agree with the phrase in lines (244-248). Where is this explained in the article? It is a serious flaw of the article.

The failure of education, possible mental problems, and learning disabilities are not specific to the LGBTQIA+ population fraction. We then enter into a public policy intervention. Rethink your approach. Finally, I do not agree with the publication of the article because it is too focused on a reality not explained in its most general scope. This is not a so good approach.

Reviewer 3 Report

I applaud the authors for making this contribution to the community and literature. There is a clear dearth of literature in the area of LGBTQ+ homelessness and pregnancy/parenting. I offer the following feedback to the authors:

1.     The abstract identifies “restorative justice” as one of the two guiding philosophies. On page 9, the manuscript states that “Community is a primary value….” And that “…this value is connected to those that come from a transformative justice approach….” Stating “restorative justice” in the abstract made this reader think that the authors would include the concept with equal weight, per se, within the program and manuscript. It’s not clear to this reader how the process of restorative justice is included in the described program.

2.     It’s unclear to this reader how youth would be recruited to the proposed housing program.

3.     Lines 284 – 287: This sentence is confusing regarding eligibility. It reads like the program is open to 1). LGBTQ+ youth, 2). Pregnant and parenting youth, and 3). LGBTQ+ youth who are pregnant and/or parenting. If this is the case then the authors should consider adding content in the manuscript that addresses participation of heterosexual, cisgender pregnant and parenting youth.

4.     Lines 540 – 545: The authors list the psychosocial outcomes to be evaluated. The program was previously described as providing funds, case management, and mental health services. Does the program include intentional components that address some of these outcomes, such as critical action and political participation? If so, should they not be described? Additionally, none of the outcomes are specific to pregnancy/parenting, which is a major component of the program.

5.     It’s unclear what supports are offered to pregnant/parenting program participants. It’s also unclear how the program addresses space for children and any supports for children themselves.

6.     In the literature review, the authors define the term ‘youth’, but they don’t specify what age range their program is designed/available for. This is important as the question of considerations for youth who are minors is not addressed.

7.     There is housing literature that suggests for programs to be effective, support that lasts for more than one year is required. The authors do not state what happens in their program to youth who might reach the 12 month participation and not be prepared/ready for independent living.

8.     Line 11: 2022 should be 2021?

9.     Line 272: extra period.

Round 2

Reviewer 2 Report

2nd Comments

In my opinion, in the current legislation on Medically Assisted Procreation (MAP) in the light of non-discrimination on grounds of gender, the concept of gender that I defend for this purpose is that contained in the Yogyakarta Principles, on the application of international legislation on human rights in relation to sexual orientation and gender identity, formulated at a conference held in Indonesia, in 2007, by a group of experts in human rights and presented to the United Nations, in Geneva, in the same year. According to these principles, gender is understood as the internal, individual and deeply felt experience that each person has in relation to gender, which may or may not correspond to the sex assigned at birth, including the personal feeling of the body (which may involve, by free choice, modification of bodily appearance or function by medical, surgical or other means) and other gender expressions, including dress, speech and mannerisms.

Within the scope of the Council of Europe, the Istanbul Convention, open for signature by the Member States of this international organization, those who are not members of it but participated in its elaboration and the European Union, in 2011, contains a definition of gender, for the purposes of of its application, centered less on the individual and subjective experience of what it means, for each person, to belong to a certain “gender”, and more on the social construction of gender in each concrete society. “Gender” thus refers to the roles, behaviors, activities and socially constructed attributes that a given society considers to be appropriate for women and men, and is considered, along with sex, an anti-discriminatory factor that must be observed in the implementation of all provisions of this international treaty. The woman is invited to be, as Gena Corea expressively writes in “The Mother Machine”, as an appearance of a new and unjust discrimination based on sex.

Another form of negative discrimination associated with techniques is discrimination based on socioeconomic status, whether Housing First or otherwise. The treatments necessary for the birth of a child through PMA are expensive and, despite the fact that infertility is classified as a disease by the World Health Organization and its treatment is assured through establishments integrated in the National Health Service, the public investment made in the area has proved to be insufficient to meet the demand for healthcare in this area in a timely manner.

These considerations are the main ones that should be revealed and explored in future analyzes and studies for the authors. As these are large when compared to the average income of the population in general, discrimination is generated here on the grounds of the socioeconomic status of infertile people: only those who have the necessary financial resources for this purpose will be able to satisfy their desire to obtain a “take-home baby”, as they are expressively named by part, above all, of the North American doctrine .

Another form of negative discrimination associated with techniques is discrimination based on socioeconomic status, whether Housing First or otherwise. The treatments necessary for the birth of a child through (MAP) are expensive and, despite the fact that infertility is classified as a disease by the World Health Organization and its treatment is assured through establishments integrated in the National Health Service, the public investment made in the area has proved to be insufficient to meet the demand for healthcare in this area in a timely manner.

Reviewer 3 Report

No comments